# Characterization of Powdered Lulo (*Solanum quitoense*) Bagasse as a Functional Food Ingredient

**DOI:** 10.3390/foods9060723

**Published:** 2020-06-02

**Authors:** Leidy Indira Hinestroza-Córdoba, Stevens Duarte Serna, Lucía Seguí, Cristina Barrera, Noelia Betoret

**Affiliations:** 1Grupo de Valoración y Aprovechamiento de la Biodiversidad, Universidad Tecnológica del Chocó, AA.292, Calle 22 No. 18B-10, Quibdó-Chocó CP. 270001, Colombia; leihicor@doctor.upv.es; 2Institute of Food Engineering for Development, Universitat Politècnica de València, CP 46022 València, Spain; steduase@etsiamn.upv.es (S.D.S.); lusegil@upv.net.upv.es (L.S.); mcbarpu@tal.upv.es (C.B.)

**Keywords:** fruit by-products, lulo bagasse powder, dehydration, fiber, antioxidant properties, carotenoids

## Abstract

The stabilization of fruit bagasse by drying and milling technology is a valuable processing technology to improve its durability and preserve its valuable biologically active components. The objective of this study was to evaluate the effect of lyophilization and air temperature (60 °C and 70 °C) in hot air-drying as well as grinding conditions (coarse or fine granulometry) on physico-chemical properties; water interaction capacity; antioxidant properties; and carotenoid content of powdered lulo bagasse. Air-drying kinetics at 60 °C and 70 °C and sorption isotherms at 20 °C were also determined. Results showed that drying conditions influence antioxidant properties and carotenoid content while granulometry slightly influenced fiber and water interaction properties. Fiber content was near 50% and carotenoid content was higher than 60 µg/g dry matter in lyophilized powder. This β-carotene content is comparable to that provided by carrot juice. Air-drying at 60 °C only reduced carotenoids content by 10%.

## 1. Introduction

Lulo (*Solanum quitoense* Lam) is one of the most important tropical fruits in Colombia. According to the report by the Ministry of Agriculture and Rural Development of Colombia (2017), Colombia is the third ranked country in Latin America with the largest number of hectares cultivated with fruit. The lulo harvest from different varieties, covering 10,539 hectares, has increased from 67,473 tons in 2012 to 79,872 tons in 2017 [1].

In recent years, lulo fruit has raised much interest in the global market due to its organoleptic characteristics, pleasant aroma, acidic and refreshing taste [2] and its high content of bioactive components such as fiber, minerals (phosphorus, calcium, iron), vitamins (thiamin, riboflavin, vitamin C), and other specific compounds (carotenes, lutein, zeaxanthin, chlorogenic acid, and bioactive amines) [3]. Moreover, a recent study has demonstrated the antihypertensive effect of compounds present in its juice and responsible for its bitter taste. These compounds are identified as bioactive amines N^1^, N^4^, N^8^-tris-(Dihydrocafeoyl), spermidine, and N^1^, N^8^-bis-(Dihydrocafeoyl) spermidine [4].

Studies about the uses of the lulo fruit in the food industry are scarce. Lulo fruit is consumed as juice, in desserts, and for jellies; processed to make frozen concentrated juice and tea infusions as well as fermented for the elaboration of alcoholic beverages. In Colombia, it is the main ingredient for making the Lulada fruit cocktail liquor such as Ecuador’s Canelazo and Colada Morada cocktail drinks. Juice production is the most frequently industrialization option. However, a large number of by-products are produced with associated environmental problems. According to data from [5], around 9.76 million tons of fruit and vegetable by-products are generated every year. Lulo bagasse includes the fruit skin as well as traces of pulp and seeds. Traditional utilization for such food industry by-products include feed for livestock, fertilizers, or agricultural substrates [6]. However, these uses do not provide adequate added value when considering the valuable active components contained in the by-products.

The stabilization of fruit bagasse by drying and milling technology is a valuable processing technology to improve its durability and preserve its biologically active components. Chemical composition and structural characteristics of raw material largely determine physico-chemical and functional properties of the final powder [7]. Powders obtained can be used as a healthy natural ingredient or as a raw material to extract bioactive compounds for other uses. The effect of drying and milling technology has been studied in fruit pomaces such as apple, grapes, cherry, blackcurrant, strawberry, raspberry, or blackberry [8]. However, as far as we know, no study has been done to date with lulo bagasse. The objective of this study was to evaluate the effect of lyophilization and air temperature (60 °C and 70 °C) in hot air-drying as well as grinding conditions (coarse or fine granulometry) on physico-chemical, water interaction, and antioxidant properties of powdered lulo bagasse. The effect on the content of the three major carotenoids has been also evaluated.

## 2. Materials and Methods

### 2.1. Lulo Bagasse Preparation

Two kg of fresh lulo fruits (*Solanum quitoense* Lam.), equivalent to 8–9 pieces, from Colombia were purchased in the Central Market in València (Spain). Whole lulo fruits were washed, blended for 10 min in a domestic blender (Phillips Avance Collection Standmixer, 800 W 2 L), and filtered with a stainless steel 500 µm sieve. After filtering, lulo juice and a solid paste were separated. The solid paste, referred to as the lulo bagasse from now on, was labeled and stored at 4 °C in a freezer until further processing.

### 2.2. Dehydration and Milling of Lulo Bagasse

Lulo bagasse was dehydrated by hot air-drying and lyophilization. Hot air-drying was carried out in a convective dryer (Pol-eko Aparatura, Katowice, Poland) at 60 °C and 70 °C until aw ≤ 0.3 was achieved. Lyophilization was performed in a lyophilizer (Telstar, Lioalta-g) at 0.05 mbar for 24 h after samples were frozen at −40 °C.

Dehydrated lulo bagasse was milled in a domestic food processor (Thermomix^®^, Vorwerk, Spain) to obtain two different granulometries (fine and coarse). Fine granulometry resulted from milling at 10,000 rpm for 2 min in 20 s intervals and the coarse one by milling at 4000 rpm for 20 s and, subsequently at 10,000 rpm for 20 s in 5 s intervals. Fine and coarse lulo bagasse powders were stored in opaque glass jars in conditions of controlled relative humidity.

### 2.3. Analytical Determinations

Water activity was measured with a dew point hygrometer (Aqualab 4TE Decagon devices Inc. Pullman, WA, USA) at the temperature of 20 °C. Moisture content was measured by drying until constant weight was achieved [9]. Total soluble solids content was determined in an ABBE ATAGO 3-T refractometer thermostated at 20 °C. Before the measurement, dried samples were water diluted in the proportion 1:10 (g/mL). Fiber content was determined following the Van Soest method as described by Mertens et al. [10] Neutral detergent fiber, acid detergent fiber, and lignin detergent fiber were analyzed and cellulose, hemicellulose, and lignin were calculated from those results.

### 2.4. Water Interaction and Emulsifying Properties

Solubility as the mass fraction of dissolved material during powder rehydration was determined following the procedure described by Mimouni et al. [11].

Hygroscopicity, defined as the capability of a product to absorb water, was evaluated according to the Cai and Corke [12] method by weighing 0.5 g of each sample and taking them to an airtight chamber next to a saturated solution of sodium sulfate (Na_2_SO_4_).

Wettability or the time taken by the powders to get completely wet was assessed by weighing 2 g of sample in a beaker with 20 mL of distilled water at 25 °C [13].

Swelling capacity is defined as the ratio between the volume that a sample occupies after hydration for a certain time and the original weight of the sample [14,15]. One gram of sample was weighed in a graduated conical tube, to which 10 mL of water was added to hydrate the sample for 18 h at 25 °C.

Water holding capacity or water bound by gravity at atmospheric pressure was determined by measuring water content of the precipitate after mixing 0.2 g of sample and 10 mL of distilled water and left to stand for 18 h at 25 °C [14].

Water retention capacity is defined as water content remaining bond after hydration and centrifugation. 1 g of sample was weighed in a conical centrifuge tube and 10 mL of water was added, allowing hydration for 18 h at 25 °C. After that, samples were centrifuged for 30 min at 514× *g*. The precipitate was weighed and lyophilized to obtain the dry weight of the sample [14].

Oil retention capacity was measured following the method described by Garau et al. [16]. A 0.2 g of sample was mixed with 1.5 g of sunflower oil at room temperature. After that, the mixture was centrifuged at 1500× *g* for 5 min, the supernatant was removed, and the precipitate was weighted. Oil retention capacity was expressed in grams of absorbed oil per gram of initial sample.

Emulsifying activity was determined by the method described by Yasumatsu et al. [17]. A 2% (*w*/*v*) aqueous solution was prepared with the sample in a graduated tube. Seven mL of the prepared solution was mixed with 7 mL of sunflower oil and homogenized for 5 min in a vortex at 2400 rpm. After that, the mix was centrifuged at 12,857× *g* for 5 min. The volume of the emulsion was measured and referred to the total fluid volume. The emulsifying stability was determined by a similar procedure explained for emulsifying activity, except that the emulsions were heated at 80 °C for 30 min before centrifugation at 514× *g* for 5 min.

### 2.5. Particle Size

Particle size was determined on wet and dry dispersions. In both cases, laser diffraction equipment (Mastersizer, Malvern Instruments Limited, Worcerster, Great Britain) was used with a short-wavelength blue light source in conjunction with forward and backscatter detection to enhance sizing performance in the range 0.01–1000 µm. For the wet measurement, a small quantity of each sample was diluted in deionized water until it reached an obscuration of 8–9%. For the dry measurement, a small amount of each sample was put directly into the equipment until it reached an obscuration of 8–9%. Particle size distribution measurements were characterized through average equivalent volume diameter D [3,4].

### 2.6. Optical Properties

CIE*L*a*b* color coordinates were determined from the surface reflectance spectra obtained between 400 and 700 nm, when measuring on white and black backgrounds, considering standard light source D65 and standard observer 10° (Minolta spectrophotometer CM-3600d, Japan).

### 2.7. Antioxidant Properties

For antioxidant extraction, the samples were mixed with an 80:20 (*v*/*v*) methanol–water solution in the proportion 1:10 (*w*/*v*) and centrifuged at 10,000 rpm for 5 min at 20 °C (Selecta, “Medrifriger BL-S”). The next analyses were carried out in the supernatant.

### 2.8. Total Phenols and Flavonoids Content

Total phenols were determined following the Folin–Ciocalteu method [18,19]. A sample of the 0.125 mL of extract, 0.125 mL of Folin–Ciocalteu reagent (Sigma-Aldrich), and 0.5 mL of double-distilled water were mixed and allowed to react for 6 min. After that, 1.25 mL of 7% (*m*/*v*) sodium carbonate solution and 1 mL of double distilled water were added. Absorbance was measured in a spectrophotometer (Thermo Scientific, Helios Zeta U/Vis) at 765 nm. A blank was used as a reference and allowed to react for 90 min. A standard gallic acid curve ranging from 0 to 500 mg/L was obtained to express results in milligrams of gallic acid equivalent (GAE) per gram of dry sample.

Flavonoid content was determined following the method described by Luximon-Ramma et al. [20] A 1.5 mL sample of extract and 1.5 mL of a 2% (*w*/*v*) aluminum chloride solution were mixed and left in the dark for 10 min. Absorbance was measured on a spectrophotometer (Thermo Scientific, Helios Zeta U/Vis) at 368 nm. The resulting data were compared to a standard quercetin curve ranging from 0 to 350 mg/L. The results were expressed in milligrams of quercetin equivalent (EQ) per gram of dry sample.

### 2.9. DPPH and ABTS Methods

Antioxidant capacity was determined following the DPPH (2,2-diphenyl-1-picrylhydrazyl) method described by Kuskoski et al. [21] and Stratil et al. [22], with some modifications. A total of 0.1 mL of the extract, 0.9 mL of methanol, and 2 mL of the methanol–DPPH solution were mixed and absorbance was measured at 517 nm in a spectrophotometer (Thermo Scientific, Helios Zeta U/Vis). The results were expressed as milligrams of Trolox equivalent (TE) per gram of dry matter, using the Trolox calibration curve within a 0 to 500 mg/L concentration range.

The antioxidant activity was also evaluated following the ABTS (2,2′-azino-bis(3-ethylbenzothiazoline-6-sulfonic acid) radical method described by Re et al. [23]. A solution of 7 mM of ABTS and 2.45 mM of potassium persulfate was prepared and left to stand in the dark at room temperature for 16 h. ABTS was mixed with phosphate buffer to reach an absorbance of 0.70 ± 0.02, read at 734 nm. A 0.1 mL of extract was added to 2.9 mL of ABTS solution and absorbance was measured at 734 nm in a spectrophotometer (Thermo Scientific, Helios Zeta UV/Vis) after 0, 3 and 7 min of reaction time. The results were expressed as mg of Trolox equivalent (TE) per gram of dry matter.

### 2.10. Carotenoid Content by HPLC (High-Performance Liquid Chromatography)

Carotenoids were extracted according to the procedure described by Rodrigues et al. [24] and Bunea et al. [25], with some modifications. One gram of sample was mixed with methanol/ethyl acetate/petroleum ether (1:1:1, *v*/*v*/*v*) as the extraction solvent. The extract was saponified for 12 h in the dark at room temperature, using a 30% (*v*/*v*) KOH solution in methanol. The sample was washed with saturated saline solution, evaporated in a rotary evaporator (T < 30 °C), and analyzed by HPLC. HPLC analysis was performed in a HPLC Alliance 2995 system, using a separation module (Waters, 2695) made up of a pump and a DAD detector (2996, Waters, Milford, MA, USA). Carotenoids were separated with a YMC C30 column [5 µm, 250 mm × 4.6 mm (internal diameter)], using ternary gradient elution made up of acetonitrile:water (9:1, *v*/*v*) with 0.25% trietylamine (solvent A) and ethyl acetate with 0.25% trietylamine (Solvent B). Carotenoids were quantified at a flow of 1 mL/min. The results were expressed in micrograms per 100 g of dry sample.

### 2.11. Sorption Isotherms

Sorption isotherms were determined according to the method described by Wolfe et al. [26] with some modifications. This technique involves the use of saturated salt solutions to maintain a known and controlled humidity environment inside a closed jar at a fixed temperature condition. One gram of sample was placed in a closed jar together with one of the next saturated salt solutions: LiCl (aw = 0.1), CH_3_COOK (aw = 0.23, MgCl_2_ (aw = 0.32), K_2_CO_3_ (aw = 0.43), Mg (NO_3_)_2_ (aw = 0.52), NaCl (aw = 0.75), KCl (aw = 0.85), and BaCl_2_ (aw = 0.90) at 20 °C. The samples were weighed every eight days until constant weight was reached. Once the samples reached equilibrium, moisture content was measured.

### 2.12. Statistical Analysis

All determinations were made in triplicate and the statistical analysis of the data was performed in a Statgraphics Centurion XVII software package, making use of a simple or multifactorial analysis of variance (ANOVA) at a 95% confidence level (*p* < 0.05).

## 3. Results

### 3.1. Hot Air-Drying of Lulo Bagasse

Figure 1 shows the hot air-drying curves at 60 °C and 70 °C and drying speed curves for lulo bagasse.

For a curve comparison, reduced humidity (X_w_/X_w0_) was used. An effect of air temperature on the kinetics of the process can be observed. Drying with air at 60 °C requires approximately 10–12 h until the product reaches an almost constant weight, which corresponds to a humidity close to 2%. Increasing the temperature to 70 °C decreased the time required to 6–7 h. Although when obtaining a functional food ingredient it is necessary to consider the effect of air-drying temperature on bioactive compounds (antioxidants such as carotenoids in lulo powder), the effect on processing time, production, and energy consumption will be relevant when it comes to setting up an industrial process. In Figure 1, it can be observed that an increase in the air-drying temperature resulted in greater values of drying rates that decreased along all the air-drying process. Experimental values were linearly adjusted to model equation showed at Table 1. Two stages with different kinetics were clearly identified. Kinetic parameters obtained for each stage at both temperatures and correlation coefficients (R^2^) are included in Table 1.

In the first stage, the decline of drying rate was not affected by air temperature. Considering that in the drying chamber the relation between air volume and the mass of lulo bagasse was large enough to avoid changes in the air conditions, the lack of temperature effect on the decline of drying rate revealed an internal control. Structural characteristics and composition determine water molecular transport from the innermost layers to the surface of the solid bed. However, the drying kinetics change when the moisture of the product drops below 40% (*w*/*w*) at 60 °C and below 30% (*w*/*w*) at 70 °C (these values are the result of considering a X_w0_ = 3.4 g water/g dry matter and the relation $\frac{\mathrm{Xw}}{{\mathrm{Xw}}_{0}}$ indicated in Table 1). From this humidity, decreasing the drying rate with moisture depends on the air temperature being more pronounced at 70 °C. Phase transitions of some components, together with a greater compaction of the lulo bagasse bed would probably explain these differences.

### 3.2. Moisture Sorption Isotherms of Lulo Powders

The sorption isotherms at 20 °C of the lulo bagasse powders are shown in Figure 2.

The curves demonstrate an important increase in water activity with low increasing equilibrium moisture content, following the type II and III BET classification shape, which is usual for non-structured and non-porous solid foods [27]. It is the typical form of plant products rich in simple sugars such as fructose or glucose and macromolecules such as cellulose or hemicellulose, with low ability to adsorb water molecules. The curves are similar to that for dried persimmon leaves [28]. Figure 2 shows the agreement between the experimental data and predicted isotherms using BET (Brunauer-Emmett-Teller) and GAB (Guggenheim-Anderson-de Boer) models. Model parameters and correlation coefficients (R^2^) are included in Table 2. The BET model was adjusted considering aw values below 0.55, which is consistent with the assumptions assumed by the model [29]. GAB model setting has included values up to 0.8.

Among all sorption isotherm model parameters, the monolayer moisture content is recognized as the most important one. It corresponds to the moisture content affording the longest storage time period with minimum quality loss by deteriorative reactions (except fat oxidation) at a given temperature. From a physico-chemical point of view, it is related to the number of sorption sites available on the material surface. Therefore, the conditions of milling or drying inducing changes in surface structural characteristics are expected to lead to changes in monolayer moisture content. Furthermore, the physical state (crystalline or amorphous) in which the food components are found and the phase transitions along the dehydration process conditioned by temperature and kinetics, strongly influence water retention.

A lower variability and greater consistency in the monolayer moisture values can be observed in Table 2, as obtained from the BET model. The values were slightly higher in the fine powder than in the coarse one, probably because the fine powder had a larger specific surface, having a greater number of accessible active points. Lyophilized powders showed lower values, which can be associated with a greater disruption of cell structure that occurs during this treatment. The greatest difference between the air-dried powders at 60 °C and 70 °C is given by the value of parameter C. It is an empirical parameter directly related to the net heat of sorption being the difference between the heat of sorption of the first layer of molecules of water and the others. It is considered that C values greater than 2 are associated with higher adsorption forces (type II sorption isotherms corresponding to dried powders at 60 °C). The differences obtained between air-dried powders at 60 °C or 70 °C could be related to the effect of temperature on the kinetics of the process and the associated phase transitions.

### 3.3. Physico-Chemical Properties

Table 3 shows the total soluble solids content (x_ss_), water activity (aw), and moisture content (x_w_) of different lulo bagasse powders. All drying conditions allow for reducing the aw below 0.27, which is considered a usual value in food powders such as milk powder or instant coffee [30]. Corresponding moisture contents were also very low, the drying process having contributed to reducing most of the free water content responsible for spoilage reactions. Statistically significant differences among treatments were detected for the three parameters, the differences being lower between fine and coarse powders. The most significant differences appeared between the two temperatures of hot air-drying. These differences manifested themselves similarly in the water activity and in the moisture content of the samples, so they are associated with the effect of temperature on the desorption characteristics of the samples determined by air-drying kinetics and phase transitions along the process.

The different fiber fractions of lulo bagasse powders determined by the Van Soest method are summarized in Table 3. This method estimates the structural carbohydrates and indigestible substances linked to them that form the plant cell wall. Although the determination quantifies the fractions of hemicellulose, cellulose, and lignin, it does not allow for the quantification of components such as pectins and other polysaccharides that are also part of the dietary fiber. It is essentially the soluble fiber that is undervalued in the Van Soest method. It can be observed that the total fiber content varied from 34% to 47% being amounts lower than the dietary fiber obtained for other fruit waste powders such as pomegranate bagasses (45.6–50.3 g/100 g dry weight) [31], grape co-products (74.5 g/100 g dry weight) [32], apple pomace (51.1 g/100 g dry weight) [33], or banana peels (49.9 g/100 g dry weight) [34]. However, the total fiber content is similar to that for other fruits by-products such as grapefruit peel (44.2%) [35], mango (37.1%), and peach (37.6%) by-products [36]. Larrauri et al. [37] stated that products containing 50% of total fiber can be considered as a source rich in fiber. According to the Scientific Opinion on Dietary Reference Values for carbohydrates and dietary fiber [38], a daily dietary intake of at least 25 g of fiber is recommended. However, nutritionists recommend a daily fiber intake of 35 g per day [39]. It can be observed that in the three drying treatments, and practically for all fiber fractions, there are statistically significant differences between fine granulometry and coarse granulometry powders. This can be explained by the effect of milling and particle size reduction. Although health claims for fiber do not distinguish between insoluble and soluble fractions, it is accepted that the ‘ideal dietary fiber’ should have a balanced composition (insoluble and soluble fractions) [37]. Other authors accept that fiber sources suitable for use as food ingredients should have a ratio soluble to insoluble close to 1:2 to provide adequate physiological effects [40]. The results obtained for the lulo bagasse powders showed a much higher relationship between hemicellulose (part of the soluble fiber) and the insoluble fiber, although as previously mentioned, the Van Soest method used does not allow for the determination of other soluble fiber components.

Figure 3 shows the particle size distribution of lulo bagasse powders by the dry and wet methods. The particle size distribution of lulo bagasse powders by the dry method did not show remarkable differences between treatments.

A particle volume maximum appeared at 1000 µm for all treatments and a minor maximum at 100 µm only for the lyophilized powders. Conversely, the dehydration treatment applied largely determined the particle size distribution by the wet method. In the powders dehydrated by hot air at 60 °C (Figure 3B) and regardless of granulometry, no large changes in distribution were observed. Only a slight increase in the number of smaller particles (around 100 µm) was observed due to the solubilization of a part of the particles sized around 1000 µm. However, in the samples dried by hot air at 70 °C and lyophilized, the granulometry largely determined the changes in the distribution obtained by wet way (Figure 3A,C). In fine granulometry powders, the changes were very similar to those obtained for drying at 60 °C, with a shift in the distribution toward smaller particle sizes. This shift was more pronounced in the powders obtained by lyophilization, probably due to the higher solubility associated with increased porosity of the particles in this treatment. However, in coarse granulometry powders, the appearance of a much more pronounced peak at a particle size slightly smaller than 1000 µm was observed, reflecting an aggregation of smaller particles (Figure 3).

Table 3 includes L*a*b* coordinates and h and C parameters of the powders. For the luminosity values (L*), a significant decrease could be observed in the air-dried powders compared with the lyophilized ones. Regarding the parameters a* and b*, which represent (+red/green−) and (+yellow/blue−) respectively, all the values were found to be positive. Both parameters were significantly lower for lyophilized powders, while the differences among the air-dried ones were slight. Among the air-dried powders, differences between fine and coarse granulometry were higher than those caused by air temperature. Formation of brown compounds as a result of the Maillard reaction in air-dried powders is favored by the major exposure to oxygen in fine powders. Lyophilization greatly reduces the Maillard reactions [41]. As a result, air-dried powders tend to have a browner color and the lyophilized ones are a greenish color more similar to the fresh bagasse.

### 3.4. Water Interaction and Emulsification Properties of Lulo Bagasse Powders

Solubility is a physical property describing powder behavior in an aqueous solution. For fiber rich ingredients such as powders from fruit by-products, it mainly depends on fiber content, size, and physical characteristics of particles (porosity and physical stage of components). The solubility of lulo bagasse powders is included in Table 4.

Values were similar to that obtained for orange peel by Garau et al. [16], but higher to that obtained for tomato powders by Santos de Sousa et al. [42]. Statistical analysis showed granulometry (fine or coarse) as the most important factor determining significant differences (*p* < 0.05) among the samples. These results are consistent with those obtained for fiber content. The greater intensity of the milling treatment applied in fine powders produces a smaller particle size, a reduction in the content of the different insoluble fiber fractions, and therefore a greater solubility. Regarding the effect of the dehydration treatment, the lyophilized samples showed higher solubility levels, as a consequence of the more aggressive structural breaks suffered by the samples due to the sublimation of the water contained within the tissue. The decrease in solubility caused by temperature increase in hot air-dried samples could be due to the degradation of pectin substances and physical changes affecting other components during the drying process. Hygroscopicity is defined as the capability of a product to absorb water [43,44], and describe that a powder with low hygroscopicity, low humidity, low caking, and high solubility can be considered stable. The results obtained showed that hygroscopicity slightly decreased as the drying temperature increased. According to Ahmed et al. [45], powder moisture variations have a direct impact on hygroscopicity. Regarding swelling capacity, water holding, and retention capacities, they depend on microstructural properties of particles and the nature of the fiber content (soluble or insoluble).

Porous particles and soluble fibers have a high water holding capacity and swelling to form viscous solutions. Insoluble fibers can also adsorb and retain water within their fibrous matrix, but to a lesser extent. Lecumberri et al. [46] provides data on the swelling and water holding capacity of cellulose (insoluble fiber) and apple and orange pectin (soluble fiber), showing undetected or practically null values in the case of cellulose. The swelling value obtained for apple pectin (7.42 ± 1.15) was similar to that obtained for the lyophilized lulo bagasse powders and it was slightly lower in the hot air-dried ones. In all lulo bagasse powders, regardless of the dehydration treatment, water holding capacity was much lower than that for apple (16.51 ± 3.77) and orange (28.07 ± 5.37) pectin, although it was very similar to the value obtained for a fiber rich cocoa product. Additionally, for this property, the values were higher in lyophilized powders. According to Lecumberri et al. [47], wettability is strongly affected by size and porosity of particles. As a consequence of water sublimation, lyophilized powders showed a more porous particle surface, which makes water difficult to penetrate due to the greater surface tension that needs to be overcome. Wettability of lyophilized lulo bagasse powders showed much higher values than the air-dried ones.

Emulsifying properties concerning oil retention capacity and emulsifying activity and stability are required to use powders as an ingredient in fatty foods. These properties require the presence of long-chain chemical compounds with hydrophilic and hydrophobic groups. An effective emulsifying agent consists of water-soluble (hydrophilic) and water-insoluble (hydrophobic) domains. Pectin has an emulsifying capacity and has been traditionally used as a gelling and thickening reagent. Recently, reports of the remarkable emulsifying ability of pectin have attracted much attention, with sugar beet pectin [48] and pomegranate peel pectin [49] as examples. Non-conclusive results were obtained for emulsifying activity and stability of the powdered lulo bagasse. Oil retention capacity was very low and may be affected by the lower pectin content of the lulo bagasse. Similar powders from lemon, orange, peach, and apple bagasse have a greater oil retention capacity (contain 2.5 to 2.9 g of oil/g of sample), according to Martínez-Las Heras et al. [50].

### 3.5. Antioxidant Properties

Total phenol and flavonoid content are represented in Figure 4.

Significant differences (*p* < 0.05) were observed between the hot air-dried powders and lyophilized ones. The low temperature and vacuum conditions in the lyophilization treatment limit oxidation reactions, thus preserving phenolic and flavonoid compounds [51] Results were similar to those reported by Crozier et al. [52] in pomegranate powder and red fruits. Some other authors have also evaluated the effect of different drying techniques (mainly hot air-drying and lyophilization) in phenol and flavonoid content of different fruit by-products such as orange by-products [16], bagasse of blackcurrant pomace [53], mango peel [54], and apple peel [55]. The results exhibited, in all cases, a decrease in antioxidant compounds as the temperature increased.

Results of the antiradical capacity by the DPPH and ABTS methods are included in Figure 4. The use of more than one single method is recommended to estimate the antioxidant activity of complex samples [56]. In this case, the ABTS free radical method, which has been reported to be more sensitive to hydrophilic antiradicals [57], was used in addition to the DPPH radical method. In coherence with total phenol and flavonoid content, both methods showed slightly higher values for lyophilized lulo bagasse powders. However, the differences were more marked for the results obtained by the ABTS radical method. Additionally, for the hydrophilic nature of antioxidants compounds in the lulo bagasse powders, differences between the DPPH and ABTS methods were probably due to their different sensitivities to the antiradical compounds that may be present in the sugarcane products. The time of reactions could also have produced some differences; nevertheless, the ABTS reaction is usually faster than the DPPH inhibition reaction [56].

### 3.6. Carotenoid Content

The β-cryptoxanthin, α-carotene and β-carotene content of lulo bagasse powders are summarized in Table 5.

A significant effect (*p* < 0.05) of dehydration treatment and air-drying temperature was observed, while the granulometry did not affect the results significantly. Lyophilized powders had the highest content in all of the components analyzed. In the hot air-dried powders, the increase in air temperature had, in all cases, a negative effect. These results highlight, once again, the relevance of lyophilized process to preserve nutraceutical components due to the low temperature and the absence of oxygen in the drying chamber [41,58] quantified β-carotene in lyophilized by-products, showing lower values for guava by-product (26.67 µg/100 g dry matter), mango (58.26 µg/100 g dry matter), or passion fruit (53.93 µg/100 g dry matter). Regarding the effect of temperature in hot air-drying, the results were consistent with those reported by Albanese et al. [59] They demonstrated hot air-drying at 50 °C as a suitable method and alternative to freeze-drying to preserve carotenoid compounds and antioxidant activity in tomato peels. Heating of the extracts up to 100 °C caused a progressive reduction of total carotenoids up to about 30% after 250 min of treatment. In lulo bagasse powders, hot air-drying at 60 °C reduced the α-carotene content by less than 10% and the β-carotene content by less than 30% compared to the lyophilized powder. Increasing the drying temperature to 70 °C increased the losses of the two components to values greater than 50%. Although the hot air-drying treatment significantly reduced the content in the analyzed components, the beta-carotene content in the lyophilized powder was similar to the carrot juice provided by Bub et al. [60] in a human intervention study (330 mL of carrot juice providing 27.1 mg of β-carotene) that demonstrated the beneficial effects of this component in reducing the oxidation of low density lipoproteins.

## 4. Conclusions

Lulo bagasse is a suitable raw material for obtaining a powder rich in fiber and carotenoids useful as an ingredient in the food industry. The imbalance in the ratio between soluble and insoluble fiber should be taken into account in subsequent applications.

Drying kinetics showed that in the first stage of the air-drying process, water molecular transport from the innermost layers to the surface of the solid bed was not affected by air temperature. Structural characteristics and the composition of bagasse determine an internal control. In this sense, the air temperature could be reduced in this stage with energy savings as a consequence.

The slight differences observed in the monolayer water content provided by isotherms for the different processing conditions, showed the expected high stability of lulo bagasse powders. Although differences in the C parameter showed important structural and physical changes along drying, physico-chemical characteristics such as color were not highly affected.

Both granulometry and dehydration conditions determine the properties of the final powder. A finer granulometry leads, independent of the dehydration conditions, to better water interaction properties and, especially to greater solubility. In relation to dehydration methods, lyophilization is the method that provides better antioxidant properties and a higher carotenoid content, although carotenoid content is also acceptable in hot air-drying at 60 °C.

## Figures and Tables

**Figure 1 foods-09-00723-f001:**
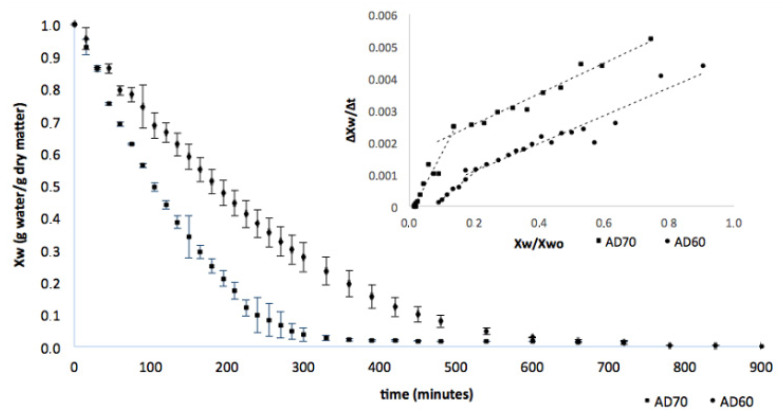
Hot air-drying and drying rate curves at 60 °C and 70 °C of lulo bagasse. AD60 and AD70: hot air-drying at 60 °C and 70 °C, respectively.

**Figure 2 foods-09-00723-f002:**
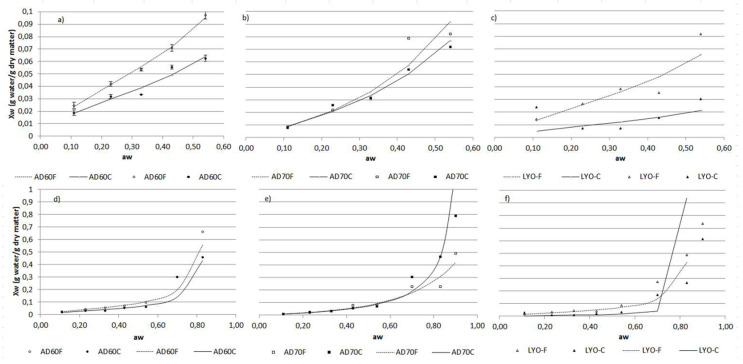
Sorption isotherms: experimental data and BET (Brunauer-Emmett-Teller) (**a**–**c**) and GAB (Guggenheim-Anderson-de Boer) (**d**–**f**) adjustment.

**Figure 3 foods-09-00723-f003:**
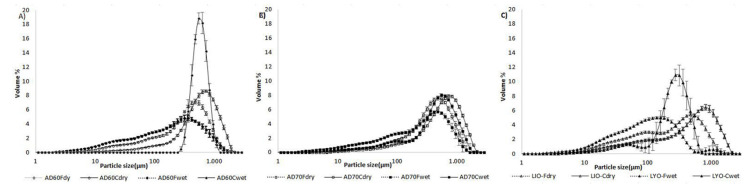
Particle size distribution of lulo bagasse powders by the dry method (empty markers) and wet method (full markers) for the three treatments considered: (**A**) air drying at 60 °C; (**B**) air drying at 70 °C; (**C**) lyophilization. AD60 and AD70: hot air-drying at 60 °C and 70 °C, respectively; LYO: lyophilized; F: fine granulometry; C: coarse granulometry.

**Figure 4 foods-09-00723-f004:**
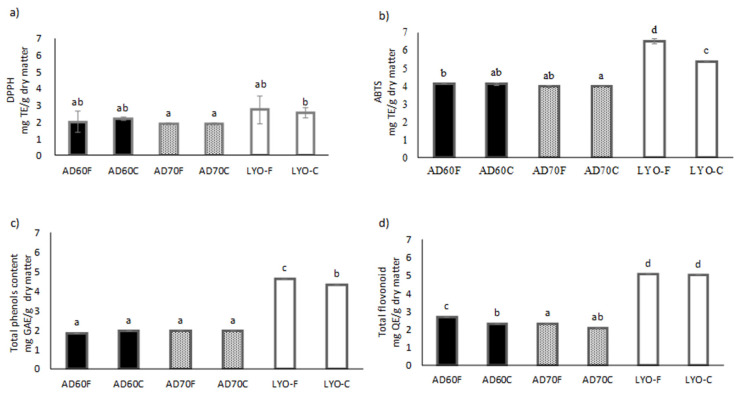
Antioxidant properties: results from DPPH method (**a**), results from ABTS method (**b**), total phenol content (**c**) and flavonoid content (**d**). AD60 and AD70: hot air-drying at 60 °C and 70 °C, respectively; LYO: lyophilized; F: fine granulometry; C: coarse granulometry.

**Table 1 foods-09-00723-t001:** Kinetics of air drying of lulo bagasse at 60 °C and 70 °C.

	60 °C	70 °C
**First stage:** $\frac{\Delta \mathrm{Xw}}{\Delta t}{=k}_{1}\frac{\mathrm{Xw}}{{\mathrm{Xw}}_{0}}{+k}_{2}$	$\frac{\mathrm{Xw}}{{\mathrm{Xw}}_{0}}\in \left[1,\;0.195\;\right]$	$\frac{\mathrm{Xw}}{{\mathrm{Xw}}_{0}}\in \left[1,\;0.129\;\right]$
k_1_	0.0043	0.0048
k_2_	0.0002	0.0016
R^2^	0.9176	0.9625
**Second stage:** $\frac{\Delta \mathrm{Xw}}{\Delta t}{=k\prime }_{1}\frac{\mathrm{Xw}}{{\mathrm{Xw}}_{0}}{+k\prime }_{2}$	$\frac{\mathrm{Xw}}{{\mathrm{Xw}}_{0}}\in \left[0.195,\;0.089\;\right]$	$\frac{\mathrm{Xw}}{{\mathrm{Xw}}_{0}}\in \left[0.129,\;0.024\;\right]$
k’_1_	0.0084	0.0195
k’_2_	−0.0006	−0.0003
R^2^	0.987	0.9319

Xw: moisture content; Xw_0_: initial moisture content.

**Table 2 foods-09-00723-t002:** Parameters from BET and GAB adjustment.

	BET [29] $\mathrm{Xw}=\frac{W_{0}{\mathrm{Ca}}_{w}}{{(1-a}_{w})(1+\left(C-1\right)a_{w})}$			GAB [26] $\mathrm{Xw}=\frac{W_{0}{\mathrm{CKa}}_{w}}{{(1-\mathrm{Ka}}_{w}{)(1-\mathrm{Ka}}_{w}{+\mathrm{CKa}}_{w})}$			
	W_0_	C	R^2^	W_0_	C	K	R^2^
AD60F	0.050	5.902	0.992	0.113	−4.103	−0.614	0.929
AD60C	0.033	8.207	0.912	0.022	0.481	−3.208	0.837
AD70F	0.089	0.768	0.058	0.056	0.897	1.299	0.918
AD70C	0.060	1.208	0.061	0.063	0.937	1.108	0.929
LYO-F	0.048	2.726	0.918	0.293	−4.611	−0.095	0.934
LYO-C	0.007	−246.4	0.619	0.056	−4.694	−0.077	0.589

AD60, AD70: hot air drying at 60 °C and 70 °C respectively; LYO: lyophilized; F: fine granulometry; C: coarse granulometry.

**Table 3 foods-09-00723-t003:** Water activity (a_w_), moisture content (x_w_), (g water/g)**,** soluble solids content (x_ss_) (g soluble solids/g), fiber content (% of dry weight) and CIE L*a*b* coordinates of lulo bagasse powders. Mean ± standard deviation of three repetitions.

	AD60F	AD60C	AD70F	AD70C	LYO-F	LYO-C
a_w_	0.119 ± 0.006 ^a^	0.199 ± 0.006 ^a^	0.258 ± 0.006 ^b^	0.267 ± 0.007 ^b^	0.166 ± 0.003 ^b^	0.134 ± 0.003 ^a^
x_w_ (g_w_/g_sample_)	0.021 ± 0.004 ^a^	0.035 ± 0.003 ^a^	0.018 ± 0.002 ^a,b^	0.015 ± 0.001 ^b,c^	0.0194 ± 0.0006 ^c^	0.022 ± 0.002 ^d^
x_ss_ (g_ss_/g_total_)	0.236 ± 0.005 ^e^	0.149 ± 0.012 ^a^	0.222 ± 0.006 ^b^	0.146 ± 0.06 ^c^	0.21 ± 0.11 ^b^	0.26 ± 0.14 ^d^
Fiber content						
Hemicellulose (%)	5.2 ± 0.2 ^a^	11.3 ± 0.2 ^e^	5.5 ± 0.2 ^a^	11.30 ± 0.4 ^c^	10.3 ± 0.2 ^b^	10.7 ± 0.4 ^b,c^
Cellulose (%)	18.6 ± 0.3 ^a^	24.7 ± 0.4 ^c^	21.3 ± 0.7 ^b^	24.59 ± 0.05 ^c^	22.1 ± 0.2 ^b^	24.6 ± 0.3 ^c^
Lignin (%)	10.1 ± 0.3 ^a^	20.84 ± 3 ^b^	17.6 ± 0.23 ^a^	10.6 ± 0.2 ^a^	8.5 ± 0.5 ^b^	10.8 ± 0.4 ^a^
Insoluble fiber (%)	35.2 ± 0.1 ^b^	40.3 ± 0.4 ^d^	33.3 ± 0.2 ^a^	34.2 ± 0.2 ^a,b^	38.4 ± 0.3 ^c^	41.65 ± 1.01 ^e^
Total fiber (%	42.6 ± 0.04 ^b^	50.6 ± 0.6 ^f^	41.4 ± 0.4 ^a^	46.5 ± 0.3 ^c^	46.0 ± 0.4 ^d^	48.1 ± 0.3 ^e^
Colour						
L*	58.4 ± 0.2 ^c^	50.8 ± 0.2 ^b^	53.3 ± 0.3 ^c^	50.97 ± 0.06 ^b^	63.134 ± 0.13 ^d^	60.59 ± 0.05 ^d^
a*	10.37 ± 0.02 ^c^	10.22 ± 0.07 ^c^	10.35 ± 0.11 ^c^	10.76 ± 0.13 ^d^	9.46 ± 0.07 ^a^	9.98 ± 0.05 ^b^
b*	38.22 ± 0.11 ^d^	40.5 ± 0.3 ^f^	39.5 ± 0.2 ^e^	38.043 ± 0.10 ^d^	36.74 ± 0.11 ^c^	34.57 ± 0.06 ^b^
C	39.60 ± 0.01 ^d^	42.6 ± 0.3 ^f^	40.8 ± 0.2 ^e^	39.53 ± 0.12 ^d^	37.94 ± 0.09 ^c^	35.98 ± 0.06 ^a^
h	74.84 ± 0.07 ^c,d^	71.92 ± 0.04 ^b^	75.32 ± 0.09 ^d^	74.2 ± 0.2 ^c^	75.56 ± 0.14 ^d^	73.70 ± 0.02 ^c^

AD60, AD70: hot air drying at 60 °C and 70 °C respectively; LYO: lyophilized; F: fine granulometry; C: coarse granulometry. ^a,b,c^ different letters on the same file indicate statistically significant differences at a 95% confidence level.

**Table 4 foods-09-00723-t004:** Hydration, water retention and emulsification properties of lulo bagasse powders. Mean ± standard deviation of three repetitions.

	AD60F	AD60C	AD70F	AD70C	LYO-F	LYO-C
Solubility (%)	35 ± 5 ^b^	27 ± 6 ^b^	32 ± 2 ^d^	19 ± 4 ^c^	45 ± 8 ^a^	30 ± 4 ^c^
Higroscopicity (g_water_/100 g)	30.9 ± 0.4 ^c^	23.0 ± 0.2 ^a^	23.0 ± 0.2 ^a^	22.7 ± 1.1 ^a^	22.1 ± 0.20 ^a^	25.32 ± 1.1 ^b^
Wettability (s)	31.7 ± 0.6 ^b^	8.7 ± 1.2 ^a^	10.0 ± 1.0 ^a^	11.0 ± 1.7 ^a^	19.5 ± 2.6 ^c^	17.0 ± 1.0 ^b^
Swelling capacity (mL_water_/g)	4.98 ± 0.02 ^b^	4.46 ± 0.04 ^a^	4.97 ± 0.02 ^b^	4.98 ± 0.05 ^b^	7.46 ± 0.05 ^d^	5.48 ± 0.02 ^c^
Water holding capacity (g_water_/g_dry matter_)	5.89 ± 0.10 ^a^	5.7 ± 0.1 ^a^	6.3 ± 0.2 ^a^	7.6 ± 0.8 ^b^	8.2 ± 0.7 ^b^	6.4 ± 0.5 ^a^
Water retention capacity (g_water_/g_dry_ _matter_)	4.75 ± 0.02 ^a^	4.5 ± 0.2 ^a^	5.5 ± 0.2 ^b^	5.83 ± 0.06 ^c,d^	5.9 ± 0.1 ^d^	5.9 ± 0.4 ^b,c^
Emulsifying properties						
Oil retention capacity (g_oil_/g_sample_)	0.142 ± 0.004 ^a^	0.18 ± 0.03 ^a,b^	0.20 ± 0.02 ^b,c^	0.20 ± 0.02 ^b^	0.24 ± 0.01 ^c^	0.45 ± 0.04 ^d^
Emulsification activity	N.D	N.D	N.D	N.D	N.D	N.D
Emulsification stability	N.D	N.D	N.D	N.D	N.D	N.D

AD60, AD70: hot air drying at 60 °C and 70 °C, respectively; LYO: lyophilized; F: fine granulometry; C: coarse granulometry. ^a,b,c^ different letters on the same column indicate statistically significant differences at a 95% confidence level. N.D: Not detected.

**Table 5 foods-09-00723-t005:** Carotenoids content (μg/g of dry sample) in the lulo bagasse powders. Mean ± standard deviation of three repetitions.

	AD60F	AD60C	AD70F	AD70C	LYO-F	LYO-C
β-cryptoxanthin	4.761 ± 0.014 ^c^	4.06 ± 0.12 ^b^	1.197 ± 0.008 ^a^	1.193 ± 0.011 ^a^	8.83 ± 0.04 ^d^	8.72 ± 0.18 ^d^
α-carotene	1.60 ± 0.07 ^b^	1.577 ± 0.013 ^b^	0.546 ± 0.017 ^a^	0.581 ± 0.015 ^a^	1.75 ± 0.01 ^c^	1.73 ± 0.03 ^c^
β-carotene	45.2 ± 0.4 ^b^	45.1 ± 0.3 ^b^	27.8 ± 0.3 ^a^	27.61 ± 0.02 ^a^	61.85 ± 0.25 ^d^	61.15 ± 0.09 ^c^
Total	51.5 ± 0.5 ^b^	50.7 ± 0.2 ^b^	29.51± 0.2 ^a^	29.38 ± 0.2 ^a^	72. 6 ± 0.4 ^d^	71.6 ± 0.3 ^c^

AD60 and AD70: Hot air-drying at 60 and 70 °C, respectively; LYO: Lyophilized; F: Fine granulometry; C: Coarse granulometry. ^a,b,c^ different letters on the same file indicate statistically significant differences at a 95% confidence level.

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
