# Peer review of "Characterization of Powdered Lulo (*Solanum quitoense*) Bagasse as a Functional Food Ingredient"

_foods, 2020, doi:10.3390/foods9060723_

Round 1

Reviewer 1 Report

The manuscript is generally very well written and contains data of some relevance for a  general readers as well as of high relevance for specialists in the topic. Although the subject of the article could be of interest for the readers of the journal, the paper needs some minor corrections.

- Introduction - not enough information about lulo.

Lulo is an interesting fruit and in my opinion the introduction should be extended to include information on the use of lulo in the food industry.

- Whole fruit or just some part was used for research?

- I think the abbreviations used in the article should be explained   e.g.: AD60, AD70, Xw/Xwo; the abbreviations in Table 2.

- Table 4 is placed in a different place from the description to this table.

- There are some editorial errors (e.g. °C, spaces):

  • Page 7, lines: 229, 244, 248  
  • Page 8, lines: 285
  • Page 9, lines: 343, 351
  • Page 10, lines: 369
  • Page 11, lines: 381
  • Page 12, lines: 414, 434

- A lot of research has been done, so in my opinion the “Conclusions” are too poor.

Reviewer 2 Report

The paper presented for review is interesting and innovative. However, I have a few questions:

How many fruits was in sample lulo fruit purchased in shop, please add this information to the chapter Material and methods, 2.1. Lulo bagasse preparation

Section 2.7.1 Total phenols and flavonoids content should be presented as separate section not sub-section

2.7. Antioxidant properties it is section about antioxidant activity measurement and should be numbered as separate section.

The results and discussion section is very well written. The authors used the latest available literature to discuss their own results.

Applications formatted correctly.

In my opinion, the manuscript presented is very interesting and brings new information in food analysis.

Reviewer 3 Report

Dear Authors,

Your idea of utilizing LULO fruit pomace by drying is really interesting, while this waste contains still a lot of  bioactive compounds. After reading your paper I have a few comments and suggestions. First of all you must submit your paper for linguistic correction, because in the current form that article is very hard to read. When you refer to literature, write the name of the first author and then: [...]. Punctuation needs to be corrected as well as the spaces between subtitles and the text. Table 4 is situated in the wrong place. Specific comments to the paper are included below.

 Abstract

Line 19 on physico-chemical …, water interaction,

what ? properties ?

Water interaction of what?

Line 21 Results show …that… drying conditions and granulometry influenced the main properties of the final powder.

The meaning of the sentence is too general and I really don’t understand what you wanted to say. Could you specify the influence?

Line 22 Drying conditions had a greater influence on antioxidant properties and carotenoid content while granulometry slightly influenced fiber and water interaction properties.
                strong?

Line 24 …powder results…?

process results in …  
powder … can be characterised by…

Abstract should be concise and encourage reading the article. Please, rewrite the Abstract.

Introduction

Line 37 …high content …of… bioactive…

Line 44 …is the most viable option… frequently used?

                Rewrite the sentence to make it understandable.

                However, lulo juice processing produces a large amount of by-product with the environmental problem…

                Why don’t you use passive form?

Line 46  According to the data from the [5] around …

Grammar!

Lines 49-50 This sentence is also not clear. Please, rewrite it.

                What is the spelling of the word valuable ? valueable?

Lines 54-55 physical-chemical and functional properties of …

                physico-chemical?

Line 57 such us or such as?

Methods

Line 68 What filter was used?
Line 69 The form will be should be replaced with has been called…?

Line 73 … until aw… was achieved

What was the pressure during lyophilization?

Line 77 Could you be more specific what does it mean “fine and coarse”?

Line 83 at the temperature of …

                Moisture content was measured by drying until the constant weight according to AOAC method [9] was achieved.

Line 85 previous? Before the…?

Line 87 …as described by the name […].

Line 87 and 88 “directly” ? This sentence is not clear.

Line 98 allowing hydration …? What does it mean?

All methods used in your research should be described by using the name of author/authors and then the citation.

Chemical compounds should be written according to the standard procedure, with subscripts.

“Swelling capacity” means the rehydration?

The text in the chapter 2.4 is very hard to understand. It should be divided into fragments/paragraphs.

Line 111 what does it mean “the mix”?

Line 113 The procedure was not the same, since the centrifugation was carried out by different conditions!!!!!!!

Line 119 …”small amount” ? How much is it?

Results

Line 190 “clear effect”  what does it mean?

I wonder if data connected with kinetics and presented in the Table 1 and Figure 1 are usually defined as MR (moisture ratio). Usually researches try to describe the drying kinetics by using popular equations (Page model, Henderson-Pabis model, Newton model etc.). Try to propose something…

Line 277 …by products…? by-products?

Line 277
[37], states that products that contain a value of 50% may be considered a source rich in fiber.

I really don’t understand what it is about.

Line 280 However, nutritionists recommend 35 g per day [39].
What do nutritionist recommend?

Line 333 Table 4 is separated from the capture. How is it possible?

Line 338 …powders showed

Conclusions

The conclusions are not relevant enough. Please rewrite them.
